# Vitrectomy combined with lens capsule flap transplantation in the treatment of high myopia macular hole retinal detachment: study protocol for a prospective randomised controlled trial

Qiaoyun Gong,[1,2,3,4,5] Luyao Ye,[1,2,3,4,5] Xia Wu,[1,2,3,4,5] Lin Xue,[1,2,3,4,5] Hao Zhou,[1,2,3,4,5] Ying Fan,[1,2,3,4,5] Xun Xu [1,2,3,4,5] Weijun Wang,[1,2,3,4,5] Tianwei Qian [1,2,3,4,5]

For numbered affiliations see end of article.

**Correspondence to**
Dr Tianwei Qian;
qtw6180@126.com

## ABSTRACT

**Introduction** Vitrectomy combined with internal limiting membrane (ILM) peeling, flap or tamponade is widely used in the treatment of macular diseases, such as macular hole (MH) and high myopia macular hole retinal detachment (HMMHRD). However, movement of the ILM to a suitable position to prevent displacement is a difficult operation. Improving visual function after surgery remains controversial. Compared with ILM, the thicker and more flexible lens capsule is easy to obtain and operate. Previous studies have confirmed the effectiveness of lens capsule flap in the treatment of MH. This study aims to evaluate the efficacy and safety of vitrectomy combined with lens capsule flap transplantation in the treatment of HMMHRD.

**Methods and analysis** This single-centre, single-blind, prospective, randomised clinical trial will include 54 patients with HMMHRD who will first undergo phacoemulsification and intraocular lens implantation and then vitrectomy combined with lens capsule flap transplantation (experimental group) or ILM tamponade (control group). Study participants will be randomly allocated in a 1:1 ratio to experimental and control groups. Follow-up will be conducted 1, 3 and 7 days and 1, 3 and 6 months after surgery in both groups. Necessary examinations will be performed at each follow-up visit. Measurement outcomes include postoperative situation of macular hole closure, best-corrected visual acuity, macular retinal function and macular retinal sensitivity. The primary outcome is type I closure rate of MH 6 months after operation. Intergroup comparisons of the proportions of patients with type I closure of MH will be performed with Fisher's exact test.

**Ethics and dissemination** Full ethics approval for this study was obtained from the Ethics Committee of Shanghai General Hospital, Shanghai Jiaotong University, Shanghai, China. The outcomes of the trial will be disseminated through peer-reviewed journals and at scientific conferences.

**Trial registration number** ChiCTR2200057836.

## STRENGTHS AND LIMITATIONS OF THIS STUDY

⇒ This is the first randomised controlled trial investigating the efficacy and safety of vitrectomy combined with lens capsule flap transplantation in the treatment of high myopia macular hole retinal detachment.

⇒ In addition to anatomical measurement, functional measurements, including best-corrected visual acuity, macular retinal function and macular retinal sensitivity, are also used to evaluate the postoperative efficacy.

⇒ Based on whether the macular hole reaches 400 µm, this trial will adopt the stratified randomisation method through the central random system.

⇒ Strict quality control measures, including adequate concealment of randomised group allocations, will be implemented.

⇒ Difference in preoperative lens opacity affecting the baseline visual acuity of the two groups and difficult evaluation of vitreoretinal surgery are the limitations of this trial.

## INTRODUCTION

High myopia is a common cause of vision loss. Globally, about 163 million individuals (2.7% of the world population) suffered from high myopia in 2000. With the increasing prevalence of HM, an estimated 1 billion individuals (9.8% of the world population) will suffer from high myopia in 2050.[1] Patients with high myopia have a higher risk of ocular disorders, such as cataract and glaucoma, than those without high myopia. In the field of the fundus, high myopia is associated with a series of pathological complications, including posterior staphyloma, myopic macular schisis (MMS), preretinal membrane, chorioretinal dystrophy, macular hole (MH) and retinal detachment (RD).[2–4] In patients with high myopia, about 6% of patients will develop MH, which is about 10 times higher than in individuals without high myopia.[5] Patients with high myopia have a long ocular axis

and a high risk of posterior scleral staphyloma. Continuous traction of the vitreous leads to the formation of MHs. The liquid in the vitreous reaches beneath the retinal neuroepithelial layer through the macular holes to form subretinal fluid (SRF), which will lead to high myopia macular hole retinal detachment (HMMHRD).[6] HMMHRD can be clearly diagnosed by optical coherence tomography (OCT). Once it occurs, vision deteriorates drastically, reducing quality of life. It is one of the serious complications threatening the vision of patients with high myopia.[7 8] Possible mechanisms of HMMHRD include vitreoretinal traction caused by epiretinal membrane, residual vitreous cortex, inner limiting membrane (ILM) and retinal vascular system,[9] which makes the vertical and tangential traction generated by the epiretinal membrane and ILM[10 11] and further weakens the adhesion between retinal neuroepithelial layer and retinal pigment epithelium (RPE). The long axis reduces the extension of the retina, resulting in the relative shortening of the retina and subsequent retinal splitting and detachment.[12]

The key to the treatment of HMMHRD is to restore the normal anatomical structure of the macular region, improve the continuous traction state of the vitreous to the retina and induce retinal reattachment. The main surgical procedures include extrascleral surgery (scleral buckling, macular banding and posterior scleral reinforcement), intraocular surgery (vitrectomy) and combined intraocular and extraocular surgery. Currently, vitreoretinal surgery is the mainstream surgical method for the treatment of HMMHRD. Among them, pars plana vitrectomy (PPV) can relieve the mechanical traction of the vitreous cortex on the macula, restore the anatomical structure, effectively reduce retinal oedema and facilitate fluid exchange, and promote the recovery of retinal function in the macular region. It is one of the most routinely performed operations.[13] However, the abnormalities of the anatomical structure in high myopia, such as long ocular axis, retinal choroidal atrophy at the posterior pole, posterior scleral staphyloma and 'macular white hole', lead to increased surgical difficulty, resulting in low retinal reduction and hole closure rate and high recurrence rate in patients with HMMHRD after simple vitrectomy. In addition, scar healing, retinal interlayer structure disorder, RPE atrophy and ganglion cell damage occur in MHs after operation, which lead to unsatisfactory visual function recovery, even if retinal anatomical reduction was achieved.[14 15] Therefore, to improve the success rate of surgery and preserve the visual function of patients as much as possible, combined macular interface surgery is needed after vitrectomy removes the traction of the vitreous to the macula. The combination of surgical methods, such as ILM peeling, ILM flap and ILM tamponade, are currently in practice. Improving the visual function of patients based on improving the reduction rate is the key to the selection of macular interface surgery for HMMHRD.[16 17]

Although these operations can improve the healing rate of MH, obtaining the ILM tissue at the edge in patients with high myopia remains challenging. The challenge is due to the problems of posterior scleral staphyloma, retinal choroidal atrophy and difficulty in identifying the ILM.[18 19] Moreover, because the ILM is originally soft, under the influence of the long axis of high myopia, ILM softens further under traction. Therefore, moving the ILM to a suitable position to prevent displacement is difficult.[19 20] As a result, the role of improving visual function postsurgery remains controversial. Therefore, the choice of operation mode of macular interface is particularly important.

In 2019, Grewal et al[21] first reported that the lens capsule was transplanted to the MH for the treatment of refractory MH. Peng et al[22 23] used an autologous or allogeneic lens capsule to treat refractory MH. The healing rate can reach 90% and 96%, and the visual acuity of patients can be improved. Similar to the ILM, the lens capsule is a kind of basement membrane. Although the lens capsule is thicker than the ILM, it is quite flexible. Therefore, positioning the lens capsule at a designated place in the retina is easier than the ILM. For most patients who need surgery for HMMHRD, the degree of lens turbidity reaches the surgical indication of cataract phacoemulsification. Therefore, most doctors will choose to complete the operation together with cataract surgery to easily obtain the capsule tissue. The mechanism of lens capsule transplantation in the treatment of MH is similar to that of ILM. The lens capsule can be used as a scaffold to promote the proliferation and migration of Müller cells in the retina and promote the healing of MHs. The study found that the lens capsule as a stent still existed and did not become thinner after nearly 3 years of follow-up.[23] Appropriate size and reasonable placement of lens capsule graft is the key to successful surgery.[23] Mastery of certain surgical skills is necessary to fill the lens capsule at the MH. Moreover, lens capsule transplantation cannot only promote the healing of MH but also improve the vision of patients.[20 24 25] In addition, lens epithelial cells on the anterior capsule flap can be removed using distilled water, resulting in low antigenicity.[23] In the actual operation of capsule transplantation, Chen et al[16 20] reported that the lens capsule is easier to process and insert MH than ILM, making it possible to vigorously promote this technology.

Based on the advantages of easy access, easy operation of lens capsule, combined with the clinical prognosis of high retinal reattachment rate and better improvement of visual function found in the capsule transplantation group in the previous study, our team plans to conduct a prospective randomised controlled trial of vitrectomy combined with lens capsule flap transplantation in the treatment of HMMHRD to explore the efficacy and safety of this operation in retinal reattachment and functional recovery after the treatment of HMMHRD.

## METHODS AND ANALYSIS
### Overview
This is a single-blind, prospective and randomised clinical trial (RCT) being conducted at Shanghai General

Hospital affiliated with Shanghai Jiaotong University. The patients have a clear diagnosis of HMMHRD and require vitrectomy. Our hypothesis is that vitrectomy combined with lens capsule flap transplantation in the treatment of HMMHRD can achieve a higher proportion of retinal reattachment and better functional recovery. After comprehensive ophthalmic and physical examinations, the patients will be randomised into an experimental group and control group to receive treatment.

## Objective

The study aims to evaluate the efficacy and safety of vitrectomy combined with lens capsule flap transplantation in retinal reattachment and functional recovery in patients with HMMHRD.

## Trial design

The prospective, single-centre RCT will enrol 54 participants that will be randomised in a 1:1 ratio to the experimental group (vitrectomy combined with lens capsule flap transplantation) or control group (vitrectomy combined with ILM tamponade). The patient's ophthalmic and systemic medical history and first examination results at admission will be recorded during the screening visit for the trial. Vitrectomy combined with lens capsule flap transplantation or ILM tamponade will be performed by one experienced surgeon. The protocol (date: 24 June 2022, version: 1.1) was developed and updated in accordance with the Consolidated Standards of Reporting Trials guidelines. This protocol was written in accordance with the Standard Protocol Items: Recommendations for Interventional Trials guidelines[26] and was approved by the Institutional Review Board of Shanghai General Hospital (online supplemental file 1), approved ID: 2022SQ284.1). The trial has been registered with the Chinese Clinical Trial Registry (ChiCTR; http://www.chictr.org.cn/, No. ChiCTR2200057836). Any protocol modifications will be communicated to the participants and trial registry. Trial results will be disseminated on trial registration (ChiCTR) or through a publication. Participants have the right to withdraw their consent and discontinue participation at any time. A flow chart of the trial design is shown in figure 1.

## Recruitment

Participants will be recruited by the surgeon during hospitalisation without any other form of advertisement. Interested patients will be invited to discuss the benefits and potential risks related to the vitrectomy combined with lens capsule flap transplantation. Participants meeting all the inclusion criteria will be fully informed of their responsibilities and all procedures involved in the trial. They will be asked to sign the written informed consent before enrolment. The English translation of informed consent can be seen in online supplemental file 2.

## Inclusion criteria

The inclusion criteria are as follows: (1) patients ≥50 years old who were diagnosed as HMMHRD by indirect ophthalmoscopy, macular OCT, fundus photography and B-ultrasound examination; (2) equivalent spherical dioptre ≤–6.00 D or axis length ≥26 mm; (3) MH as main cause of RD; (4) turbid lens without obvious calcification in the lens capsule, requiring combination with cataract surgery; (5) the range of RD within the retinal vascular arch of the posterior pole; and (6) consent to undergo surgical treatment and regular follow-up and sign relevant informed consent.

## Exclusion criteria

Exclusion criteria are as follows: (1) patients with RD with other types of holes except MH, choroidal detachment, obvious proliferative vitreoretinopathy, intraocular haemorrhage, severe cataract, glaucoma, retinal vascular inflammation or obstruction, macular degeneration, macular neovascularisation, macular haemorrhage, pathological myopia and macular atrophy; (2) patients with previous history of intraocular surgery (including PPV and intravitreal drug injection); (3) patients whose general condition cannot tolerate the operation; (4) patients who cannot cooperate with the examination and follow-up for <6 months; (5) patients unwilling to undergo concurrent vitrectomy and cataract surgery; (6) patients with concomitant diseases or conditions of the target eye or whole body (including malignant hypertension; AIDS; malignant tumour; serious mental, cardiovascular, neurological, respiratory, digestive or other systemic disease; long-term use of hormones; and immunodeficiency after heart stenting or organ transplantation); (7) patients with diabetes with poor blood glucose control; (8) patients with poor communication skills and compliance; (9) patients without the ability to follow the postoperative position; and (10) participation in other trials.

## Withdrawal criteria

Withdrawal criteria are as follows: (1) loss to follow-up; (2) withdrawal of informed consent; (3) participation in other clinical studies during the study; (4) emergence of other eye diseases during the study; (5) need of other eye treatments during the study; (6) decision by researchers to withdraw participant due to safety concerns; and (7) decision by researchers to withdraw patient due to unsuitability of treatment methods (including remedial treatment) for patient.

## Randomisation and blinding

The stratified randomisation method will be adopted in allocating participants to the two groups in a 1:1 ratio using the central random system. The stratification factors are as follows: (1) MH size >400 μm and (2) MH size ≤400 μm. Except the surgeon, all researchers, including the outcome assessors, statisticians and data analysts, will be blinded to group assignment, but providers of the intervention/control will be informed by necessity. Vitrectomy combined with lens capsule flap transplantation or ILM tamponade will be performed by one experienced

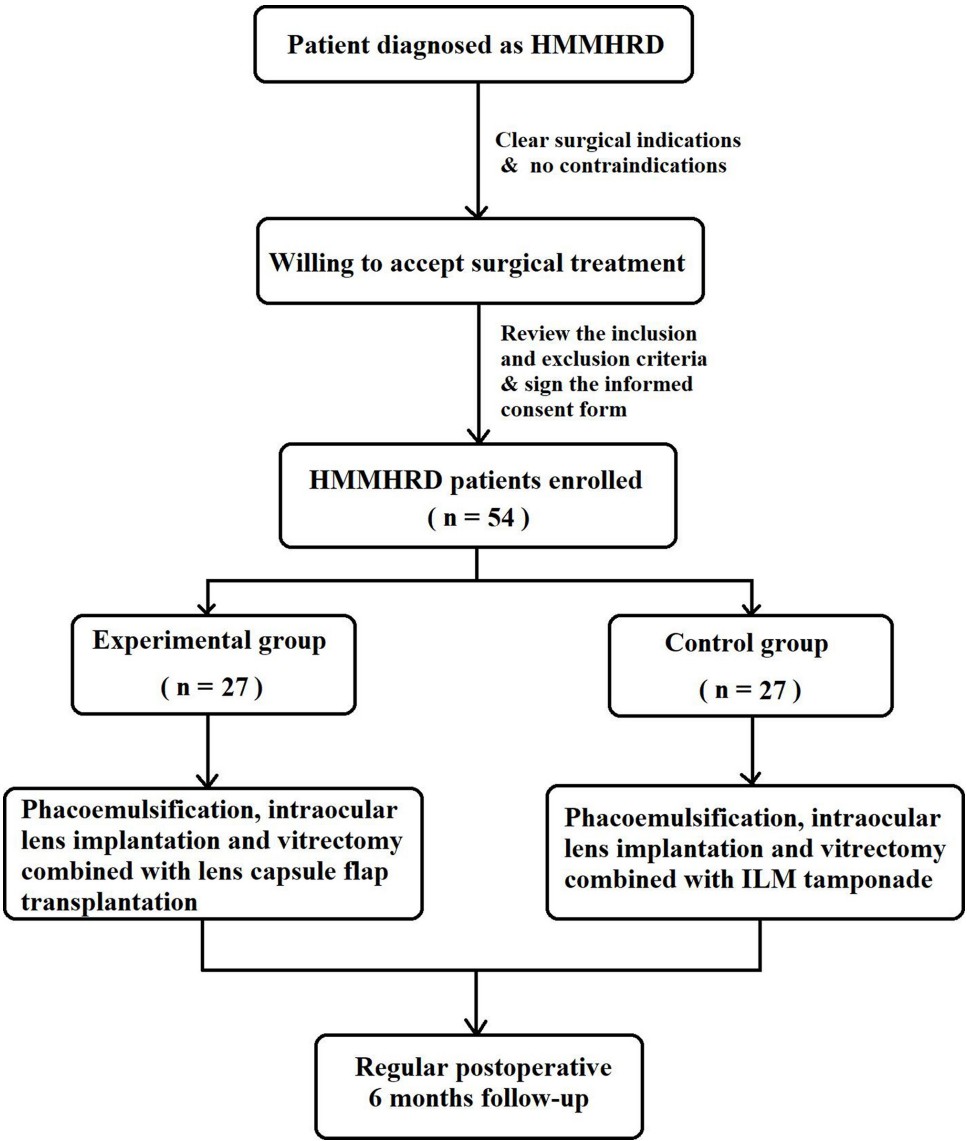

**Figure 1** Flow chart of the trial design. ILM, inner limiting membrane.

surgeon. Before the trial, researchers will be fully trained in the randomisation procedure and made aware of their individual responsibilities. The successful implementation and maintenance of the randomisation and blinding method will be validated.

## Interventions

All participants who meet the inclusion criteria of this study and do not meet any of the exclusion criteria can be included in this study. Before the surgery, the patients should receive complete preoperative ophthalmic examination, including best corrected visual acuity (BCVA), slit lamp inspection, axial length, presence of staphyloma, myopia dioptres, intraocular pressure (IOP), ocular B-ultrasound, OCT, multifocal electrophysiology (mfERG), MP-1 microperimeter, fluorescence fundus angiography and indocyanine green angiography. After passing the screening, the participants who provided written informed consent will be randomly assigned to two parallel trial groups in a 1:1 ratio according to the central

randomisation system. The patients will undergo vitrectomy combined with lens capsule flap transplantation or ILM tamponade and regular postsurgery follow-up. Participants will be blinded to the specific surgical methods. The timeline of data collection is shown in table 1.

## Experimental group

Patients will first undergo phacoemulsification and intraocular lens implantation. During this operation, the lens capsule flap will be obtained and placed in sterile distilled water for at least 30 min. Thereafter, the 23-gauge vitrectomy through the pars plana will be performed, and the ILM will be stained with brilliant blue. After brilliant blue staining, the ILM in the macular area will be completely removed with intraocular forceps. Then the small piece of lens capsule flap is moved into the vitreous cavity with microforceps trough the 23-gauge trocar. The trimming of lens capsule flap will be performed in the vitreous cavity, and the lens capsule flap can be directly placed above the MH surface. The capsular flap should be

**Table 1** Outcome measurements and data collection at each follow-up timepoint

| Phase | | Screen | Preoperative examination | Postoperative follow-up | | | | | |
|---|---|---|---|---|---|---|---|---|---|
| Follow-up | | 1 | 2 | 3 | 4 | 5 | 6 | 7 | 14 |
| Time points | | Within 1 week before operation | One day before operation | 1 day after operation | 3 days after operation | 7 days after operation | 1 month after operation | 3 months after operation | 6 months after operation |
| Informed consent | | ● | | | | | | | |
| Inclusion and exclusion criteria | | ● | | | | | | | |
| Withdrawal criteria | | | ● | ● | ● | ● | ● | ● | ● |
| Medical history collection | | ● | | | | | | | |
| Ophthalmic examination | BCVA | ● | ● | ● | ● | ● | ● | ● | ● |
| | Slit lamp inspection | ● | ● | ● | ● | ● | ● | ● | ● |
| | Axial length | ● | | | | | | ● | ● |
| | Presence of staphyloma | ● | | | | | | ● | ● |
| | Myopia dioptres | ● | | | | | | ● | ● |
| | IOP | ● | ● | ● | ● | ● | ● | ● | ● |
| | Ocular B-ultrasound | ● | ● | ● | ● | ● | ● | ● | ● |
| | OCT | ● | ● | ● | ● | ● | ● | ● | ● |
| | mfERG | ● | | | | | | ● | ● |
| | MP-1 microperimeter | ● | | | | | | ● | ● |
| | FFA and ICGA | ● | | | | | | | ● |
| Systemic examination | Vital signs | ● | ● | ● | ● | ● | ● | ● | ● |
| | Physical examination | ● | | | | | | | ● |
| Laboratory examination | routine blood test | ● | ● | | | | ● | | ● |
| | Blood biochemistry | ● | | | | | ● | | ● |
| | Coagulation function | ● | | | | | | | ● |
| | Urinalysis | ● | | | | | | | ● |
| | Immune and infection measurements | ● | | | | ● | ● | | ● |
| | ECG | ● | | | | | | | ● |
| | Chest CT | ● | | | | | | | ● |
| Adverse events recorded | | ● | ● | ● | ● | ● | ● | ● | ● |

BCVA, best corrected visual acuity; IOP, intraocular pressure; OCT, optical coherence tomography; mfERG, multifocal electrophysiology; FFA, fluorescence fundus angiography; ICGA, indocyanine green angiography.

trimmed by vitrectomy cutter as round as possible, and its diameter is about twice that of the MH. Afterwards, the lens capsule flap will need to be inserted into the MH to complete gas–liquid exchange. In this process, the operator will ensure that the capsule flap is always right under the MH. Finally, silicone oil (Oxane 5700 Bausch & Lomb, Kingston-upon-Thames, UK) will be injected into the vitreous cavity.

## Control group

Patients will first undergo phacoemulsification and intraocular lens implantation. Thereafter, 23-gauge vitrectomy through the pars plana will be performed. After complete vitreous removal, ILM will be stained with brilliant blue. After brilliant blue staining, the ILM of the retina in the brilliant blue staining area will be directly clamped, and the ILM ring in the area will be removed about 1.5-disc

diameters away from the MH. Care will be taken to prevent peeling off the ILM valve at the edge of the MH. Thereafter, the free ILM flap will be turned over and filled into the MH. During the operation, accurate filling of ILM into the MH will be confirmed severally. After complete gas–liquid exchange, silicone oil (Oxane 5700 Bausch & Lomb) will be injected into the vitreous cavity.

## Postoperative management
All patients will be administered with anti-inflammatory and anti-infective treatment after operation. During hospitalisation, IOP will be monitored two to three times daily. If IOP increases, it will be reduced. In case it is not controlled, mannitol will be administered intravenously to control IOP. If necessary, anterior chamber puncture, aspiration and drainage or glaucoma surgery will be performed.

## Outcome measurements
Follow-up will be conducted at 1, 3 and 7 days and 1, 3 and 6 months after surgery in both groups. Necessary examinations will be performed at each follow-up visit.

### Anatomical measurement
OCT will be used to assess the retina and macula before and after surgery. According to the postoperative situation of retinal reattachment and MH closure, the surgical result will be classified as: (1) type I closure: the two ends of the MH are closely connected and close to the RPE layer without any obvious exposure of the RPE layer and subretinal fluid (SRF); (2) type II closure: the two ends of the MH are not connected, and the edge of the hole is closely attached to the RPE layer with a distance between the two ends. The exposed RPE layer can be observed without any obvious SRF; (3) unclosed: the two ends of the MH are not connected, and the edge of the hole is still raised and tilted away from the RPE layer. The SRF can be observed further.[27 28]

### Functional measurement
Postoperative visual acuity, macular retinal function and macular retinal sensitivity are all important functional outcomes. (1) Central vision: BCVA; (2) macular retinal function: mfERG collects the retinal amplitude information caused by specific stimulation, forms a three-dimensional topographic map and collects P1 wave response density data for intuitive evaluation of macular retinal function; (3) macular retinal sensitivity: MP-1 microperimetry is used to examine the microvisual field of the macula.

## Primary outcome
The primary outcome is type I closure rate of MH 6 months after operation.

## Secondary outcomes
The following indicators will be used to comprehensively assess anatomical outcome: (1) type I closure rate of MH 3 months after operation; (2) type II closure rate of MH 6 months after operation; and (3) type II closure rate of MH 3 months after operation.

Functional outcomes will be assessed as follows: (1) changes in BCVA 3 and 6 months after operation compared with baseline; (2) changes in P1 wave response density in the macular region 3 and 6 months after operation compared with baseline; and (3) changes in macular retinal sensitivity 3 and 6 months after operation compared with baseline.

Furthermore, the number of adverse events and reactions, abnormality of ophthalmic examination, abnormal systemic symptoms and laboratory results will be recorded at each follow-up visit.

## Data management and safety monitoring board
Basic demographic information and data on history of ocular diseases will be collected preintervention. All the data in this study will be collected from the clinical information system of our hospital and recorded in the case report format (CRF) in electric data collection system developed by Shanghai General Hospital. Every participant included will be distinguished with a study identifier and initial without full name. An electronic password-protected document containing all the information from the CRFs will be set up for statistical analysis. Relevant documents will be stored at the Department of Ophthalmology of Shanghai General Hospital after the study. The confidentiality and safety of participants' data are guaranteed. All the data will be reserved for 5 years for the purpose of the designated scientific study.

To ensure the integrity of the RCT and protect the rights and health of the participants, a Data and Safety Monitoring Board will be constituted. As an independent advisory group, it will ensure the scientific integrity and ethical standards of the RCT and be responsible for data evaluation during the study period. The Data and Safety Monitoring Board will be developed in accordance with the Operational Guidelines for the Establishment and Functioning of Data and Safety Monitoring Boards of the WHO. Researchers will be trained to collect good quality data, promote participant retention and complete follow-up. Data analysts will be trained on data entry, coding, security and storage. Statisticians will provide training on data assessment and statistical analysis.

## Safety and combined operation-related adverse events
Possible phacoemulsification and intraocular lens implantation combined with vitrectomy complications include intraocular haemorrhage, suprachoroidal haemorrhage, endophthalmitis, uveitis, intraocular lens dislocation and corneal endothelial decompensation. Lens capsule flap transplantation may increase the risk of rejection, although the capsule flap will be immersed in sterile distilled water for 30 min. However, to date, no adverse reactions caused by lens capsule flap transplantation for the treatment of MH have been reported. During the trial, we will record and analyse any surgery-related adverse events (including the type and number) in each group

and possible causes. Patients will receive appropriate intervention for any adverse events that occur during the treatment and follow-up phases. Severe adverse events will be reported immediately to the primary investigator, and affected participants will be withdrawn from the study.

## Patients and public involvement

Neither the patients nor the public will be involved in the design of the study, nor will they be involved in its conduct (including outcome measurements). No attempt will be made to assess the burden of the intervention on the patients by themselves. Results excluding personal data will be made available on request.

## Sample size

Although no RCT of lens capsule flap transplantation in the treatment of HMMHRD has been reported, according to the existing published papers[23 29] related to this study, the type I closure rate in the experimental group was expected to be 96.0%, whereas the rate in the control group was should be 65.0%. An a priori power analysis conducted using a two-tailed test (power=0.8) at the 5% significance level indicated that 21 participants are required in each group. Considering the attrition rate of 20%, the sample size in each group was determined as 27. With the 1:1 allocation, 54 participants will be recruited.

## Statistical analysis

SPSS (V.22.0; SPSS, Inc) and GraphPad Prism (V.8, GraphPad software Inc, San Diego, California, USA) will be used for data processing and analyses. Categorical variables will be expressed as numbers and frequencies, and between-group differences will be evaluated with Fisher's exact test. Continuous variables will be expressed as the mean±SD. The Kolmogorov-Smirnov test will be used to determine whether continuous variables are normally distributed. The Student's t-test will be used for normally distributed data, and the Wilcoxon signed-rank test will be used for non-normally distributed data. All statistical tests will be two-sided with $p < 0.05$ considered statistically significant.

## ETHICS AND DISSEMINATION

This RCT has been designed in accordance with the principles of the Declaration of Helsinki. The trial protocol has been approved by the Shanghai General Hospital Institutional Review Board (Approval ID: 2022SQ284.1). It is registered on the primary registry of the WHO registry network (Chinese Clinical Trial Registry). Signed consent will be obtained from each patient after they have been informed of the study procedures, possible risks and their right to withdraw from the trial. The results of the trial without personal data will be directly communicated to participants and to the public through peer-reviewed publications and conference presentations. Standard and on-site screening are being conducted by an experienced

retinal surgeon. Anaesthesiologists will monitor the whole surgical process.

## Trial status

This study is currently in the preparatory stage. The first participant will be enrolled in April 2023. The projected completion date is March 2025.

## Author affiliations
[1]Department of Ophthalmology, Shanghai General Hospital, Shanghai, China
[2]National Clinical Research Center for Eye Diseases, Shanghai, People's Republic of China
[3]Shanghai Key Laboratory of Ocular Fundus Diseases, Shanghai, People's Republic of China
[4]Shanghai Engineering Center for Visual Science and Photomedicine, Shanghai, People's Republic of China
[5]Shanghai Engineering Center for Precise Diagnosis and Treatment of Eye Disease, Shanghai, People's Republic of China

**Acknowledgements** The trial will be conducted at Shanghai General Hospital. The authors would like to thank the patients and their family members, as well as the nurses, pharmacy staff, fellows, statisticians and clinicians for making this study possible.

**Contributors** TQ, QG, LY, YF, WW and XX participated in trial design. TQ, QG, LX, YF and WW designed the data analysis plan. LY, XW, HZ and WW collected the information needed to conduct this trial. QG, WW and TQ wrote the first draft of the manuscript. All authors reviewed and revised the manuscript and gave final approval for the publication of this study protocol.

**Funding** This study was supported by Special Project for Clinical Research in Health Industry of Shanghai Health Commission (No. 20214Y0045).

**Competing interests** None declared.

**Patient and public involvement** Patients and/or the public were not involved in the design, or conduct, or reporting, or dissemination plans of this research.

**Patient consent for publication** Consent obtained directly from patient(s)

**Ethics approval** The study protocol has been approved by the Shanghai General Hospital Institutional Review Board (Approved ID: 2022SQ284.1). Participants gave informed consent to participate in the study before taking part.

**Provenance and peer review** Not commissioned; externally peer reviewed.

### ORCID iDs
Xun Xu http://orcid.org/0000-0002-4246-4343
Tianwei Qian http://orcid.org/0000-0002-7359-911X

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
