## [Reviewer comments · BMJ Open]

ARTICLE DETAILS

TITLE (PROVISIONAL)	Vitrectomy combined with lens capsule flap transplantation in the treatment of high myopia macular hole retinal detachment: study protocol for a prospective randomized controlled trial
AUTHORS	Gong, Qiaoyun; Ye, Luyao; Wu, Xia; Xue, Lin; Zhou, Hao; Fan, Ying; Xu, Xun; Wang, Weijun; Qian, Tianwei

VERSION 1 – REVIEW

REVIEWER	Filipe Sousa Neves Centro Hospitalar de Vila Nova de Gaia Espinho EPE
REVIEW RETURNED	24-May-2022

GENERAL COMMENTS	Dear Editor in Chief and Authors Firstly, this article could be an addition to the comprehension of the treatment of high myopic macular hole retinal detachments. The main subject of the article is interesting and may should be addressed with this rigorous study design. All randomized trials are very welcome, particularly in this specific topic. However, I found some important issues that need to be addressed. From my point of view, the article reach the higher quality parameters needed in order to be published in an indexed journal. Comments: 1. The main topic of this article is very relevant and this study design is suitable to its purpose. However, vitreoretinal surgery is difficult to evaluate, as all patients may be very different from each other. Having said that, the effort employed in this study is of most importance.2. It is important to describe in both groups: pre-operative axial length, presence of staphyloma, myopia dioptres, etc. These patients have a very poor visual prognosis, not only due to retinal detachment and macular hole, but also from the previous foveal status, as the authors may find amblyopic eyes, foveal disruption of the external retina, etc. One might find difficult to differentiate the efficacy of the surgery based on visual acuity or other visual function parameters. However, as I have already said, all efforts are worthy.4. It is very difficult to measure a macular hole in a detached retina, and don't think that it is an easy thing to do in all cases. Do you have intraoperative OCT?6. Sometimes the best choice for an eye with a high myopic retinal detachment with a macular hole is a macular indentation. Do you plan to do any of these when a posterior staphyloma is present?7. Why not include younger patients?8. If a patient does not meet inclusion criteria that should be stated in the protocol, as the authors may find some difficulties in having could OCT images.
--

	9. Excluding patients during the study is not an exclusion criteria. 10. Please specify what it is “an appropriate size of the lens capsule flap” 11. Indocyanine green has been associated with retinal and neurologic toxicity. Would you consider using brilliant blue for ILM staining? 12. The capsule flap will be inserted inside the retina or above the retina? 13. Please be careful with the dates employed, when will the study start and finish? The authors wrote two different dates, in the methods and discussion section. 14. When was the silicone removed? Are these results under silicone oil? Silicone should be removed before final visit as BCVA and macular hole closure rate are only accurate after silicone oil removal.
--	--

REVIEWER	Masood Bagheri Kermanshah University of Medical Sciences, Department of Ophthalmology, Imam Khomeini Eye Center, Kermanshah University of Medical Sciences, Kermanshah, Iran.
REVIEW RETURNED	26-May-2022

GENERAL COMMENTS	Greetings and respect; Dear researchers with thanks for your good study, the following corrections are recommended:  1. Replace references 5, 12, and 30 with newer references. 2. Why pathological myopia is one of the exclusion criteria and pathological or high myopic criteria are based on patient refractive error or axial length? 3. Will patients without the ability to follow the post operation position be excluded from the study? 4. What is the rationale for using silicone oil instead of intraocular gas endotamponade? (This article can help you: Bagheri M, Najafi A, Eftekhari Milani A, Hazeri S. Efficacy of new emerging surgical approaches for macular hole retinal detachment in myopic patients; a systematic review. Expert Review of Ophthalmology. 2021 Apr 29:1-0) 5. What kind of silicon is used (Hd, 5700, 1300, etc)? Sincerely, Dr. Bagheri M.
--

VERSION 1 – AUTHOR RESPONSE

Reviewer 1

1. The main topic of this article is very relevant and this study design is suitable to its purpose. However, vitreoretinal surgery is difficult to evaluate, as all patients may be very different from each other. Having said that, the effort employed in this study is of most importance.

Author response: Thank you very much for your positive comments. Indeed, vitreoretinal surgery is difficult to evaluate. The situation during the operation is ever-changing, and it is difficult to be consistent, because all patients may be very different. This is also a limitation of this study and I added it to the strengths and limitations of this study section.

2. It is important to describe in both groups: pre-operative axial length, presence of staphyloma, myopia dioptres, etc. These patients have a very poor visual prognosis, not only due to retinal detachment and macular hole, but also from the previous foveal status, as the authors may find amblyopic eyes, foveal disruption of the external retina, etc. One might find difficult to differentiate the efficacy of the surgery based on visual acuity or other visual function parameters. However, as I have already said, all efforts are worthy.

Author response: Thank you very much and this is a very valuable comment. We added descriptions of pre-operative axial length, presence of staphyloma and myopia dioptres and paid attention to them during the postoperative follow-up. The revision can be seen in Table 1. As the reviewer mentioned, these patients have a very poor visual prognosis, not only due to retinal detachment and macular hole, but also from the previous foveal status. We quite agree with the reviewer. Therefore, the primary outcome of this RCT is Type I closure rate of MH 6 months after operation, which belongs anatomical outcome. For the patients with HMMHRD, we need to pay more attention to anatomical recovery.

4. It is very difficult to measure a macular hole in a detached retina, and don't think that it is an easy thing to do in all cases. Do you have intraoperative OCT?

Author response: Thank you very much. We quite agree with the reviewer and it is indeed very difficult to measure a macular hole in a detached retina. We have intraoperative OCT and we will perform surgery under the guidance of intraoperative OCT.

6. Sometimes the best choice for an eye with a high myopic retinal detachment with a macular hole is a macular indentation. Do you plan to do any of these when a posterior staphyloma is present?

Author response: Thank you very much and this is a very valuable comment. Macular indentation is the best choice for the patients with HMMHRD, especially a posterior staphyloma is present. Before the surgery, we will tell the patients the differences between the two surgical methods and their respective advantages according to the patient's lens opacity and the position relationship between posterior staphyloma and macula. The choice of specific operation method is decided by the patient himself/herself. If the patients choose macular indentation, we will respect patients' opinions. In receiving macular indentation, the patient will not be included in this RCT.

7. Why not include younger patients?

Author response: Thank you very much. All the patients included in this study will first undergo phacoemulsification and intraocular lens implantation no matter what in the experimental group or the

control group. And the lens capsule flap will be used in next surgery in the experimental group. That is, the patients should have turbid lens and require combination with cataract surgery. Therefore, younger patients will not be included in this study.

8. If a patient does not meet inclusion criteria that should be stated in the protocol, as the authors may find some difficulties in having could OCT images.

Author response: Thank you very much and this is a very valuable comment. OCT images is very important in this study and the diagnosis of HMMHRD is inseparable from OCT images. That is to say, patients cannot be included in this study without OCT images.

9. Excluding patients during the study is not an exclusion criteria.

Author response: Thank you very much. We removed the statement in exclusion criteria section.

10. Please specify what it is “an appropriate size of the lens capsule flap”

Author response: Thank you very much and this is a very valuable and important comment. The trimming of lens capsule flap was performed in the vitreous cavity, and the lens capsule flap can be directly placed above the MH surface. The capsular flap should be trimmed as round as possible, and its diameter is about twice that of the macular hole. We specified it and optimized surgical sequence in the protocol.

11. Indocyanine green has been associated with retinal and neurologic toxicity. Would you consider using brilliant blue for ILM staining?

Author response: Thank you very much and we quite agree with you. We will use brilliant blue for ILM staining. We revised it in the protocol.

12. The capsule flap will be inserted inside the retina or above the retina?

Author response: Thank you very much. The capsule flap should be inserted inside the retina. According to the reference [23], insertion of the lens capsular flap into the macular hole has a positive effect on the closure of macular hole.

13. Please be careful with the dates employed, when will the study start and finish? The authors wrote two different dates, in the methods and discussion section.

Author response: Thank you for your comment. I am very sorry that I made an imprecise statement. The correct description should be: "The first participant will be enrolled in April 2023. The projected completion date is March 2025." I corrected it the protocol.

14. When was the silicone removed? Are these results under silicone oil? Silicone should be removed before final visit as BCVA and macular hole closure rate are only accurate after silicone oil removal.

Author response: Thank you very much and this is a very valuable and important comment. We quite agree with you that BCVA and macular hole closure rate are only accurate after silicone oil removal. We will remove the silicone oil about 5 months after the operation to get accurate BCVA and macular hole closure rate.

Reviewer 2

1. Replace references 5, 12, and 30 with newer references.

Author response: Thank you very much. The original references 5 and 12 has been replaced with newer references. Reference 30 was deleted along with the discussion section.

2. Why pathological myopia is one of the exclusion criteria and pathological or high myopic criteria are based on patient refractive error or axial length?

Author response: Thank you very much for your comment. High myopia is usually defined as a spherical equivalent of ≤ -6.0 diopters or an ocular axial length ≥ 26.0 or 26.5 mm (Ruiz-Medrano J, Montero JA, Flores-Moreno I, Arias L, García-Layana A, Ruiz-Moreno JM. Myopic maculopathy: Current status and proposal for a new classification and grading system (ATN). *Prog Retin Eye Res.* 2019 Mar;69:80-115.). Although the terms high myopia and pathological myopia are often used interchangeably, they do not refer to the same eye disease. Pathological myopia may have many characteristic findings, including retinal, choroidal, and scleral thinning, chorioretinal atrophy and choroidal neovascularization. Pathological myopia is a highly complex disease and its pathological changes are also complex and changeable, which makes it difficult for the baseline levels of the two groups to be consistent, and also increases the uncertainty of the operation. Therefore, pathological myopia is one of the exclusion criteria.

3. Will patients without the ability to follow the post operation position be excluded from the study?

Author response: Thank you for your comment and we quite agree with you. I am very sorry that the exclusion criteria were not fully described. I have revised my description to make the exclusion criteria clearer. The updated protocol and patient consent form were approved by the Institutional Review Board of Shanghai General Hospital. The new approved ID is 2022SQ284.1

4. What is the rationale for using silicone oil instead of intraocular gas endotamponade? (This article can help you: Bagheri M, Najafi A, Eftekhari Milani A, Hazeri S. Efficacy of new emerging surgical approaches for macular hole retinal detachment in myopic patients; a systematic review. *Expert Review of Ophthalmology*. 2021 Apr 29:1-0)

Author response: Thank you for your comment. This is a very valuable and helpful comment. In this article mentioned by the reviewer, the authors drew the conclusion that perfluoropropane (C₃F₈) and silicone oil are preferred over sulfur-hexafluoride (SF₆). In patients with HMMHRD, because of the atrophic RPE and lack of retinobulbar coagulation maneuver around the hole, it is critical to support the retina for a long period to restore the integrity between the RPE and neurosensory retina. Although there is no statistically significant difference, the anatomical success rate was better in the application of the silicone oil than C₃F₈. Especially for severe visual disturbances in fellow eyes (such as elderly cases), silicone oil is recommended as a choice. Furthermore, according to the actual clinical situation in China, silicone oil is more widely used than C₃F₈. Considering the above reasons, we chose silicone oil finally.

5. What kind of silicon is used (Hd, 5700, 1300, etc)?

Author response: Thank you very much for your comment. The silicon used is Oxane 5700. We added the type of silicone oil in the protocol.

Point-by-point responses to the comments have been listed in this response letter. On behalf of my co-authors, I would like to declare that each of the changes made to this protocol in the revision and the way his or her name is listed has been seen and agreed on by each of the coauthors. Authors' institutional affiliation and the corresponding author's full address, email address has been listed in the title page.

We thank you for this opportunity to revise our protocol and to resubmit it for consideration of publication in the *BMJ Open*.

VERSION 2 – REVIEW

REVIEWER	Filipe Sousa Neves Centro Hospitalar de Vila Nova de Gaia Espinho EPE
REVIEW RETURNED	17-Jul-2022

GENERAL COMMENTS	All comments mentioned were reviewed accordingly.
---

REVIEWER	Masood Bagheri Kermanshah University of Medical Sciences, Department of Ophthalmology, Imam Khomeini Eye Center, Kermanshah University of Medical Sciences, Kermanshah, Iran.
REVIEW RETURNED	07-Jul-2022

GENERAL COMMENTS	Thanks for the corrections. with the best wishes.
--